# SUMO Modification of OsFKBP20-1b Is Integral to Proper Pre-mRNA Splicing upon Heat Stress in Rice

**DOI:** 10.3390/ijms22169049

**Published:** 2021-08-22

**Authors:** Hyun-Ji Park, Hae-Myeong Jung, Areum Lee, Seung-Hee Jo, Hyo-Jun Lee, Hyun-Soon Kim, Choon-Kyun Jung, Sung-Ran Min, Hye-Sun Cho

**Affiliations:** 1Plant Systems Engineering Research Center, Korea Research Institute of Bioscience and Biotechnology (KRIBB), Daejeon 34141, Korea; hjpark@kribb.re.kr (H.-J.P.); hmjung@kribb.re.kr (H.-M.J.); lar1027@kribb.re.kr (A.L.); chohee0720@kribb.re.kr (S.-H.J.); hyojunlee@kribb.re.kr (H.-J.L.); hyuns@kribb.re.kr (H.-S.K.); 2Department of Biosystems and Bioengineering, KRIBB School of Biotechnology, Korea University of Science and Technology, Daejeon 34113, Korea; 3Department of Functional Genomics, KRIBB School of Biotechnology, Korea University of Science and Technology, Daejeon 34113, Korea; 4Department of International Agricultural Technology and Crop Biotechnology Institute/Green Bio Science and Technology, Seoul National University, Pyeongchang 25354, Korea; jasmin@snu.ac.kr; 5Department of Agriculture, Forestry, and Bioresources and Integrated Major in Global Smart Farm, College of Agriculture and Life Sciences, Seoul National University, Seoul 08826, Korea

**Keywords:** SUMOylation, OsFKBP20-1b, splicing factor, heat stress, pre-mRNA splicing

## Abstract

OsFKBP20-1b, a plant-specific cyclophilin protein, has been implicated to regulate pre-mRNA splicing under stress conditions in rice. Here, we demonstrated that OsFKBP20-1b is SUMOylated in a reconstituted SUMOylation system in *E.coli* and in planta, and that the SUMOylation-coupled regulation was associated with enhanced protein stability using a less SUMOylated OsFKBP20-1b mutant (5KR_OsFKBP20-1b). Furthermore, OsFKBP20-1b directly interacted with OsSUMO1 and OsSUMO2 in the nucleus and cytoplasm, whereas the less SUMOylated 5KR_OsFKBP20-1b mutant had an impaired interaction with OsSUMO1 and 2 in the cytoplasm but not in the nucleus. Under heat stress, the abundance of an OsFKBP20-1b-GFP fusion protein was substantially increased in the nuclear speckles and cytoplasmic foci, whereas the heat-responsiveness was remarkably diminished in the presence of the less SUMOylated 5KR_OsFKBP20-1b-GFP mutant. The accumulation of endogenous SUMOylated OsFKBP20-1b was enhanced by heat stress in planta. Moreover, 5KR_OsFKBP20-1b was not sufficiently associated with the *U* *snRNAs* in the nucleus as a spliceosome component. A protoplast transfection assay indicated that the low SUMOylation level of 5KR_OsFKBP20-1b led to inaccurate alternative splicing and transcription under heat stress. Thus, our results suggest that OsFKBP20-1b is post-translationally regulated by SUMOylation, and the modification is crucial for proper RNA processing in response to heat stress in rice.

## 1. Introduction

Eukaryotic cells coordinate the regulation of transcriptional, translational, and posttranslational processes to cope with living under frequently changing and unfavorable environmental conditions [1]. Especially, plants as relatively stationary organisms have evolved to surmount stress-induced changes by employing various defense mechanisms, such as stress tolerance, plasticity, or recovery capacity [2,3,4,5]. Although stressor-induced transcriptional regulation is a key mechanism in cellular adaptation, posttranslational modification (PTM) of proteins also contributes to quickly controlling cellular stress responses by altering preexisting proteins [6,7]. PTM-related mechanisms include chemical alterations by reversible covalent modifications, allosteric or non-allosteric enzyme modulations, and protein structure changes induced by external stimuli [1], all of which expand protein function in an extensive network of cellular processes [8,9,10]. There are over 300 different types of PTMs in eukaryotic cells, and new ones are regularly discovered [11]. The PTM mechanisms involved in plant stress response have been extensively explored [12,13,14,15,16].

One fundamental reversible PTM mechanism is the covalent attachment of a small ubiquitin-related modifier (SUMO) to intracellular substrate proteins in plant stress responses [17,18]. Although SUMO and ubiquitin proteins have only limited sequence identity, the three-dimensional structure of SUMO proteins, which are found in all eukaryotes, resembles that of ubiquitin. Conjugation of SUMO, which occurs in a manner analogous to the enzymatic cascade for ubiquitination, is catalyzed by a set of well-conserved enzymes [19,20]. In contrast to ubiquitination, SUMOylation is typically not associated with protein degradation by the 26S proteasome. SUMO addition or elimination to intracellular proteins can modulate protein stability, enzyme activity, subcellular localization, protein-protein interactions, and transcriptional regulation [21,22]. Upon induction by various stressors, most eukaryotic cells prevalently accumulate SUMO-conjugates linked via the consensus motif, ψKXE/D, in the substrate proteins [17,23,24]. Recent plant proteomic studies focused on the SUMOylome identified over a thousand targets under stress conditions [25,26,27]. Moreover, mutational studies of the SUMO machinery have indicated that SUMOylation is an essential protective mechanism employed by plants against stress [28,29,30,31].

Global quantification studies of SUMO targets in both plants and humans indicate that a significant proportion of SUMOylated proteins function as factors regulating RNA processing and metabolism [27,32,33]. Expanding upon the mechanistic observations, a growing body of biological evidence strongly supports that SUMO modification is crucial for many aspects of RNA metabolism [34,35,36]. An analysis that combined the results from 22 proteomic studies found that the largest SUMOylated cluster corresponded to proteins involved in pre-mRNA splicing, followed by ribosomal proteins and factors involved in ribosome biogenesis [33]. High enrichment for SUMOylated proteins has identified many spliceosome components, and further analysis has associated SUMOylation to various spliceosome structure rearrangements that modulate the specificity and efficiency of RNA splicing [37,38]. For instance, the serine/arginine-rich (SR) splicing factor 1 (SRSF1) functions as a PTM regulator involving SUMOylation because its expression affects the SUMO-conjugate formation of other spliceosome proteins, including U2AF65, Snu114, Prp28, and Prp3 [39]. In a Prp3 SUMOylation-deficient mutant, the interaction of Prp3 with snRNP components U2 and U5 was impaired, suggesting that Prp3 SUMOylation is required for the proper assembly of U4/U5 and U6 tri-snRNP to form active spliceosome complexes [38]. Nevertheless, not much is known about the biological role of splicing factor SUMOylation.

Recent research found that Rice OsFKBP20-1b interacts with OsSR45 and prevents 26S proteasome-dependent degradation of the protein, identifying it as a spliceosome component that functions in RNA splicing [40]. An earlier in vitro study indicated that the rice protein FKBP20, a paralog of OsFKBP20-b, directly interacted with the SUMO-conjugating enzyme (SCE) [41]. However, the biological function of FKBP20, along with its SUMOylation status, remains unknown. In this study, we report the conjugation of SUMO to OsFKBP20-1b in a reconstituted SUMOylation system and in plant cells. We found that SUMOylation is essential for the stability of OsFKBP20-1b and facilitates its proper subcellular localization in the nucleus and cytoplasm. OsFKBP20-1b is involved in spliceosome assembly and affects alternatively splicing regulation upon heat stress.

## 2. Results

### 2.1. OsFKBP20-1b Is a SUMOylation Substrate in a Reconstituted E. coli System

A previous report indicated that OsFKBP20 (formerly known as OsFKBP20-1a) interacts with rice OsSCE1 SUMO-conjugating enzymes, OsSCE1 and OsSCE2, based on the yeast two-hybrid assay [41]. In the present study, we investigated potential interactions between the other paralog OsFKBP20-1b and the OsSCE proteins. We initially examined the subcellular localization of the rice OsSCEs by creating fusion constructs consisting of GFP and the three rice SCE genes, OsSCE1, OsSCE2, and OsSCE3. The analysis confirmed that all three C-terminal GFP fusion OsSCE (OsSCE1, 2, 3-GFP) proteins were localized in both the nucleoplasm and cytoplasm (Figure 1a). Then, each OsSCE was N-terminally fused to the YFP fragment (OsSCE1-nEYFP, OsSCE2-nEYFP, and OsSCE3-nEYFP), and OsFKBP20-1b was C-terminally fused to the YFP fragment (OsFKBP20-1b-cEYFP). Tobacco leaves were co-infiltrated with the new constructs. We detected strong fluorescence signals in the nucleus and cytoplasm of tobacco cells co-expressing OsFKBP20-1b-cEYFP and the three OsSCE-nEYFP. Interestingly, substantial interactions between two proteins in the nuclear bodies are clearly indicated by the bright fluorescence signals shown in the enlarged image sections, in contrast to the control cells without interactions (Figure 1b). These results are consistent with the previously reported intracellular localization of OsFKBP20-1b [40], and suggest that OsFKBP20-1b can specifically bind to the OsSCE proteins in the nucleus and cytoplasm.

Our findings demonstrated that OsFKBP20-1b interacted with OsSCEs, which suggested that this protein could potentially be a substrate of SUMOylation. By analyzing putative SUMOylation sites of OsFKBP20-1b protein using the SUMOplotTM program (http://www.abgent.com/doc/sumoplot (accessed on 14 March 2020), we identified three consensus and two non-consensus SUMOyaltion sites. Three lysines (K22, K152, and K173) were predicted to be potential SUMOylation sites with high scores (0.79) in OsFKBP20-1b, while K81 and K177 were predicted as putative SUMOylated sites with low scores, 0.63 and 0.62, respectively. In addition, a putative SUMO-interacting motif (SIM) was predicted as a 5-amino acid stretch from the 34th to 38th amino acid (IVDVH) using the GPS-SUMO online program (http://sumosp.biocuckoo.org (accessed on 14 March 2020)) (Appendix A). Thus, we conducted the reconstituted SUMOylation assay as reported by Okada et al. [42]. The HA-tagged OsFKBP20-1b protein was expressed in *E. coli* cells, which expressed all components for SUMOylation, including SAE1a, SAE2 (E1), SCE1a (E2), and AtSUMO variants (AtSUMO1/3-AA/GG). The immunoblotting analysis detected SUMOylated OsFKBP20-1b (~40 kDa) only in combination with the conjugatable AtSUMO1/3-GG (◀), but not with the non-conjugatable AtSUMO1/3-AA. The non-SUMOylated OsFKBP20-1b band (~25 kDa) was present in all lanes (indicated by ◁). The molecular weight of the second OsFKBP20-1b band was shifted by approximately 15 kDa due to SUMOylation, indicating that this protein was likely to be mono-SUMOylated in vitro (Appendix A). Thus, we demonstrated that OsFKBP20-1b is post-translationally regulated by SUMOylation in vitro.

Furthermore, to determine the SUMOylated lysine residues in OsFKBP20-1b, we introduced point mutations (K to R) into the potential SUMOylation sites, generating single (K22R/K81R/ K152R/ K173R/ K177R), double (K22,152R/K22,173R/K152,173R), triple (3KR; K22,152,173R), and quintuple (5KR; K22,81,152,173,177R) mutants, along with the sim mutant (IVDV to AADA) (Appendix A). We found that even the 5KR mutant still retained a certain level of SUMOylation, although the modification was substantially decreased approximately 50% (Appendix A). This result indicated that the other lysine residues could function as minor SUMOylation sites. Thus, the nuclear-cytoplasmic OsFKBP20-1b protein appears to be SUMOylated with single moieties at multiple lysine residues. However, the sim mutant did not affect its SUMOylation.

### 2.2. 5KR_OsFKBP20-1b Diminishes Association with SUMOs in the Cytoplasm

We explored whether the reduced SUMOylation of OsFKBP20-1b (5KR_OsFKBP20-1b) affect the physiological properties of the protein. We initially assessed the interaction between the OsSUMO proteins (fused to cEYFP) and WT_ or 5KR_OsFKBP20-1b (fused to nEYEP) using the BiFC assay. By alignment analysis, two paralogues of rice SUMO (OsSUMO1, and OsSUMO2) were found [43]. Unexpectedly, WT_OsFKB20-1b interacted with OsSUMO1 and OsSUMO2 in the cytoplasm and nucleus, whereas 5KR_OsFKBP20-1b interacted with these proteins only in the nucleus but not in the cytoplasm (Figure 2a). We analyzed the relative GFP signal ratio of the nuclei and cytoplasm using the ZEN system of the confocal microscope. The GFP signal intensity of WT_OsFKBP20-1b interacting with OsSUMO1 or OsSUMO2 was unevenly distributed, with ~75% detected in the nucleus and ~25% in the cytoplasm. However, the 5KR_OsFKBP20-1b mutant had less than 1% of the GFP signal intensity in the cytoplasm (Figure 2b). These results indicated that OsFKBP20-1b interacted with OsSUMO proteins in the cytoplasm via a few strong SUMOylation sites.

### 2.3. SUMO Modification Stabilizes the OsFKBP20-1b Protein in the Reconstituted E. coli System and in Planta

To assess the effect of the SUMO modification on the stability of OsFKBP20-1b, we measured the protein degradation rates of the recombinant, HA-tag-linked OsFKBP20-1b protein with or without SUMOylation whose expression underwent the reconstituted *E. coli* system using plant extracts as a protease source. The recombinant OsFKBP20-1b proteins were purified using anti-HA magnetic beads and incubated with the protein extracts prepared from young rice seedlings. The non-SUMOylated OsFKBP20-1b protein was substantially degraded in a time-dependent manner compared to the SUMOylated OsFKBP20-1b protein. However, when the reaction mixture was treated with MG132 to block the 26S proteasome activity, neither the SUMOylated nor the non-SUMOylated OsFKBP20-1b protein was not degraded throughout the time course experiment (Figure 3a). The degradation ratios indicated that non-SUMOylated OsFKBP20-1b was more rapidly degraded than SUMOylated OsFKBP20-1b in plant extracts. Approximately 50% of the initial non-SUMOylated OsFKBP20-1b protein was degraded after 120 min, whereas 75% of the initial SUMOylated OsFMBP20-1b protein remained at the last time point (Figure 3b). The results suggested that SUMOylation increases the stability of recombinant OsFKBP20-1b protein by preventing proteasome-mediated protein degradation.

To monitor the role of OsFKBP20-1b SUMOylation, a cycloheximide (CHX) assay was performed to compare the protein stability between WT_OsFKBP20-1b and the 5KR mutant in *N. benthamiana*. Before analyzing the protein levels, we confirmed that the transcript levels of OsFKBP20-1b did not differ between WT and the 5KR mutant (Appendix A). The plant-expressed recombinant proteins, WT_OsFKBP20-1b and 5KR_OsFKBP20-1b, had two immunoreactive bands detected by the anti-HA antibody, one corresponding to the predicted size of the unmodified OsFKBP20-1b (pink arrow; bottom) and the other to the modified OsFKBP20-1b (blue arrow; upper band). Both of 5KR_OsFKBP20-1b bands were more degraded under CHX treatment in proportion to the processing time than the WT bands. The upper OsFKBP20-1b band of the 5KR_OsFKBP20-1b mutant was more expressed than the corresponding WT band in *N. benthamiana* (Figure 3c). Next, we calculated and compared the relative amounts of the two recombinant OsFKBP20-1b constructs. Under CHX treatment, the stability of the upper modified OsFKBP20-1b band was approximately 2-fold lower in the 5KR mutant than in the WT construct, and the modified mutant construct was partially rescued by proteasome inhibitor MG132, but not the WT construct. The unmodified OsFKBP20-1b was highly unstable under CHX treatment; its stability was approximately 5-fold lower in the 5KR mutant than in the WT, and the degradation of the unmodified mutant was almost 2-fold reduced by MG132, but not the unmodified WT (Figure 3d). This result revealed that SUMO modification is crucial for stabilizing OsFKBP20-1b in planta. 

### 2.4. Low SUMO Modification of OsFKBP20-1b Limits the Protein to Nuclear Speckles and Cytoplasmic Foci under Heat Stress

We monitored the localization of the WT_ and 5KR_OsFKBP20-1b proteins in the absence or presence of heat stress (42 °C for 1 h) to investigate whether low SUMOylation of OsFKBP20-1b is associated mis-localization in plant cells. Under normal conditions, WT_OsFKBP20-1b-GFP was found in nuclear bodies and cytoplasmic foci, as previously reported. However, 5KR_OsFKBP20-1b-GFP was more abundantly localized in the cytoplasmic foci than the WT form (about 2-fold), but the localization to the nuclear bodies did not differ substantially between the mutant and the WT. In contrast, the less SUMOylated 5KR_OsFKBP20-1b mutant did virtually not respond to heat stress in the cytoplasmic distribution, which was quite different from the WT (Figure 4a). We expanded the analysis of the intra-nuclear localization under heat stress. We found that WT_OsFKBP20-1b-GFP responded to the heat stress with the formation of numerous nuclear speckles, whereas the localization of the less SUMOylated mutant protein did not significantly respond to heat stress with nuclear speckles (Figure 4b). We quantified the number of localizations of WT_ and 5KR_OsFKBP20-1b in both nuclear speckles and cytoplasmic foci with or without heat stress (Figure 4c). The number of cytoplasmic foci per cell of WT_OsFKBP20-1b was increased by approximately 30-fold under heat stress compared to the normal conditions. However, the 5KR mutant had an almost 1.5-fold increased number of cytoplasmic foci as heat stress response. Similarly, the number of nuclear speckles was approximately increased 6-fold in response to heat stress in the WT, whereas there was no significant change in the 5KR mutant. We also performed a cell fractionation assay to biochemically confirm the results of intracellular localization under heat stress. From the results we suggest that heat stress greatly affects translocalization of OsFKBP20-1b among intersubcellualr organelles, nucleus, and cytosol.

In many cases, SUMOylation is likely to regulate protein–protein interactions [44]. Here we performed the BiFC assay using the less SUMOylated 5KR_OsFKBP20-1b in combination with OsSR45 and OsU1-70K because it was previously reported that OsSR45 interacts with OsFKBP20-1 [40], and U1-70K interacts with SR45 [45]. As shown in Appendix A, the BiFC signals were detected between 5KR_OsFKBP20-1b and OsSR45 in both nuclear speckles and cytoplasmic foci, and it also interacted with OsU1-70K in the nuclear speckles. In a comparison of the numbers of interacting positive cells per unit area, there was no difference between the WT and the 5KR mutant (Appendix A). Then, we further monitored the cytoplasmic foci localization because it did not respond in the 5KR_OsFKBP20-1b mutant, unlike the WT, under heat stress. We tested the co-localization of GFP fused to WT_OsFKBP20-1b and 5KR_OsFKBP20-1b with the red fluorescent protein (RFP) fused AtDCP1 and AtUBP1 representing marker proteins of the P-body and stress granule (SG), receptively. We found that the P-body marker AtDCP1 was co-localized with WT_ and 5KR_OsFKBP20-1b, mainly forming a large body around the nucleus (Appendix A). However, for SG marker AtUBP1, the co-localization with WT_OsFKBP20-1b was prominently increased in the cytoplasm foci under heat stress, whereas 5KR_OsFKBP20-1b was infrequently co-localized with AtUBP1 under heat stress (Figure 4d). A graphical comparison of the co-localization of OsFKBP20-1bs and AtUBP1 was used to derive R-score values. The co-localization of the less SUMOylated 5KR mutant was significantly reduced than that of the WT_OsFKBP20-1b under heat stress (Figure 4e). The results indicated that OsFKBP20-1b is precisely localized in the intra-nuclear and cytoplasmic foci mediated by SUMO modification, especially in response to heat stress.

### 2.5. OsFKBP20-1b Is SUMOylated by Heat Stress

SUMO conjugates noticeably increase when plant cells are exposed to environmental stresses, in particular, SUMO 1/2 conjugation is an early plant response to heat stress [46]. Since OsFKBP20-1b is conjugated to SUMO1 and SUMO3 in reconstitutive *E. coli* cells, we tested endogenous OsFKBP20-1b SUMOylation status in response to heat stress. To exclude the effect of newly synthesized OsFKBP20-1b, we performed cyclohexamide treatment before the time course heat stress treatment (42 °C for 0, 3, 6, and 10 h). The WT_ and 5KR_OsFKBP20-1b proteins were mainly detected to 35 kDa size band from the total proteins in the normal condition, while, these proteins exhibited multiple slowly migrating bands above the original band (>60 kDa, SUMO conjugated-like) within 3 h of heat treatment (Figure 5a). Interestingly, the SUMO conjugated-like WT_OsFKBP20-1b distinctly increased, and this increase maintained from 3 h and until 10 h after heat treatment. We the quantified the SUMO conjugated-like bands of OsFKBP20-1b. Compared to the band intensity of WT_OsFKBP20-1b, that of the SUMO conjugated-like protein bands of 5KR_OsFKBP20-1b decreased by approximately 3- and 2-fold at 3 and 10 h, respectively, and decreased a little at 6 h (Figure 5b). Next we determined whether OsFKBP20-1b is a SUMO substrate in planta by performing IP with anti-Flag under heat stress conditions (42 °C for 3 h). Before the IP step, all SUMOylated proteins were detected by the anti-SUMO1 antibody, whereas only SUMOylated OsFKBP20-1b proteins were detectable after the IP step. Subsequent immunoblot analysis of the pull-down proteins revealed that substantially fewer 5KR_OsFKBP20-1b were SUMOylated than WT_OsFKBP20-1b and OsFKBP20-1b was SUMOylated in planta (Figure 5c). From the result, we suggest the SUMO conjugation is relatively reduced in the 5KR SUMOylation site mutant.

### 2.6. SUMO Modification of OsFKBP20-1b Is Crucial for Sustaining Proper RNA Processing in Response to Heat Stress

We previously reported that OsFKBP20-1b participates in the RNA splicing process by regulating OsSR45 protein stability [40]. Here, we investigated whether the SUMOylation of OsFKBP20-1b affects OsSR45 protein stability. The protoplast transfection assay was performed in *Osfkbp20-1b* knock out (*k*/*o*) mutant. The OsSR45-2HA protein was co-expressed with WT_ or 5KR_OsFKBP20-1b 3 × FLAG fusion proteins. Consistent with the previous results, the stability of OsSR45 protein was substantially decreased in the *k*/*o* cells compared with the WT cells, whereas the stability was maintained when the recombinant WT_OsFKBP20-1b was co-expressed. Interestingly, the less SUMOylated 5KR_OsFKBP20-1b mutant protein could also rescue the OsSR45 stability with the same effect as WT_OsFKBP20-1b (Appendix A).

Because OsFKBP20-1b functions in the RNA splicing, its SUMOylation status may affect the association of the spliceosome components. This possibility was examined using the RNA-immunoprecipitation (RIP) assay. The WT_ and 5KR_OsFKBP20-1 GFP-fusion proteins were transiently expressed in *N. benthamiana* leaves using GFP co-expression as a negative control. After separating the nuclear protein by fractionation, IP was performed to isolate the GFP fusion proteins using a GFP trap (Figure 6a). Then, the expression levels of *U1*, *U2*, *U4*, *U5*, and *U6* small nuclear RNA (snRNA), which are core components of each spliceosome complex, were assessed by RT-qPCR. We found that the expression of all snRNAs was higher in the presence of WT_OsFKBP20-1b, especially for *U4*, *U5*, and U6 tri-snRNPs, than in the GFP negative control. In contrast, the values for *U* snRNAs (*U4*, *U5*, and *U6*) association were prominently lower for the 5KR_OsFKBP20-1b mutant than for the WT (Figure 6b).

We then assessed whether the less SUMOylated OsFKBP20-1b directly affected RNA processing under heat stress. We performed the protoplast transfection for WT_ and 5KR_OsFKBP20-1b in *osfkbp20-1b k*/*o* seedlings and analyzed the transcript levels of stress-responsive genes (*OsLEA3* and *OsNAC5*) and heat shock protein genes (*OsHSP101* and *OsHSP90*). Notably, the expression levels of *OsHSP101* and *OsHSP90* were increased by WT_OsFKBP20-1b without stress treatment. In addition, after 30 min of heat stress, the expression levels of *OsLEA3*, *OsNAC5*, and *OsHSP101* were approximately 1.5–2-fold increased by complementation of WT_OsFKBP20-1b compared with those in the *osfkbp20-1b k*/*o* control, whereas the expression values of these genes were less increased by 5KR_OsFKBP20-1b, except *OsHSP90* (Figure 6c). Moreover, to confirm the pre-mRNA splicing efficiency associated with the SUMOylation of OsFKBP20-1b, we examined the splicing patterns of the *OsHsfA2a*, *OsHsf2d*, and *OsLea3* genes after heat stress treatment. The three genes generated alternative splicing (AS) isoforms induced by heat stress, as shown in Figure 6d. *OsHsfA2a* is one of the best-characterized genes that produce AS isoforms under heat stress. The isoform 2 of *OsHsfA2a* was decreased in WT_OsFKBP20-1b, but the 5KR mutant maintained the same level as the *k*/*o* negative control after 60 min of heat stress. The isoform 2 expression of OsHsf2d was increased under heat stress, but this increase was less detectable in 5KR_OsFKBP20-1 than in the WT_OsFKBP20-1b. Another stress-responsive gene, *OsLea3*, generated the isoform 2 under various stress conditions. The expression of *OsLea3* isoform 2 was increased after 30 min heat stress and maintained until 60 min in WT_OsFKBP20-1b, whereas the 5KR mutant-expressing protoplasts indicated that the expression was more increased without heat stress; however, it was inversely reduced at 60 min compared to the *k*/*o* control and the WT_OsFKBP20-1b recombinant expression (Figure 6e). To determine whether the phenological response of OsFKBP20-1b is influenced by heat stress, we applied heat stress to 3-week-old *osfkbp20-1b* and the WT plants at 42 °C for 30 h, with subsequent recovery under normal conditions (at 28 °C) for 4 days. *osfkbp20-1b k*/*o* plants showed a hypersensitive phenotype to heat stress compared to that in WT both following heat stress and recovery. Meanwhile, before heat treatment, there was no difference between *osfkbp20-1b* and WT (Appendix A). Hence, we propose that SUMO modification of OsFKBP20-1b is required for the association with snRNPs within spliceosome complexes and the proper RNA processing of stress response genes upon heat stress.

## 3. Discussion

Our previous study has established that a novel plant-specific immunophilin OsFKBP20-1b interacts with the SR protein SR45 and regulates the protein stability as a PTM regulator. Moreover, upon stress conditions, OsFKBP20-1b functions in the RNA processing of a subset of stress response genes [40]. In this report, we extend this model by showing that OsFKBP20-1b is also subjected to PTM and provide evidence that the SUMO modification of OsFKBP20-1b is required for proper RNA splicing under heat stress in rice. 

It is not surprising that OsFKBP20-1b protein is modulated by SUMO because a previous Y2H assay showed that OsSCE1 physically interacts with the OsFKBP20 protein [41]. Consistent with the earlier report, our BiFC analysis revealed that OsFKBP20-1b, a paralog of OsFKBP20 protein, was associated with all three OsSCEs proteins in plant cells (Figure 1b). Interestingly, the localization of two of the OsSCEs proteins is associated with the nuclear bodies, implying the possibility that the modification of the SUMO substrate OsFKBP20-1b occurs in the nuclear body, which harbors the SUMO units and the SUMO modification machinery, or it recruits SCE1 into the nuclear bodies. In general, the reconstituted SUMOylation system in *E. coli* cells lacks SUMO-specific protease [47], which allows for excessive lysine residue SUMOylation in recombinant eukaryotic proteins using this system [48,49]. In the present study, we did not yet identify the distinct SUMO target sites of OsFKBP20-1b, but we confirmed the OsFKBP20-1b is a SUMOylated protein and identified several key SUMO binding sites for AtSUMO1 at five lysine residues using the reconstituted *E. coli* system (Appendix A). Although the less SUMOylated mutant was not perfect for inspecting SUMO regulation mechanism of OsFKBP20-1b protein in planta, it roughly represents known SUMO modification effects in cellular conditions (Figure 2a). In contrast to the reconstituted SUMOylation system, the endogenous SUMO protein was difficult to detect or isolate from plant cells because this modification is extremely transient [50]. Furthermore, the endogenous forms of SUMO-conjugated OsFKBP20-1b protein could be visualized only after immunoprecipitation of OsFKBP20-1b protein, and the less SUMOylated protein, 5KR_OsFKBP20-1b, had a prominently diminished SUMO conjugation under heat stress in planta (Figure 5c). Our data indicate that OsFKBP20-1b is SUMOylated at multiple lysine residues, including the five predicted lysine residues, by the complex SUMOylation system in plants. 

In the present work, we observed that the hypo-SUMOylated OsFKBP20-1b protein was decayed more quickly than WT under protease-supplied conditions (Figure 3). Previous studies have shown that SUMOylation promotes protein stability. For example, SUMOylation is required for DREB2A protein stability [51] along with the stability of RACK1B, IRF4, and GPS2 [52,53,54]. However, it is also known that SUMOylation often negatively regulates protein stability. For instance, deficient SUMOylation of HOXA10 protein is associated with enhanced protein stability, and the SUMO modification is targeted by the DNA-binding subunit of Dbf4-dependent kinase proteasomal degradation [55,56]. Our data add support to the observation that SUMO modification prevents protein degradation mediated by the proteasome complex, and, therefore, it confers protein stability. 

We found that the SUMOylation-mediated, proper localization of OsFKBP20-1b in nuclear speckles and cytoplasmic foci supported the previously reported SUMO mechanism (Figure 4). RanGAP, the first identified SUMO substrate, is translocated into the nuclear pore where it is SUMOylated by interacting with a SUMO E3 ligase RanBP2 [57]. Moreover, it has been demonstrated that the subcellular localization of essential proteins involved in cellular signaling depends on the SUMOylation of their lysine residues [22,58,59]. Here, we addressed the molecular function of the SUMO modification in OsFKBP20-1b by focusing on the pre-mRNA splicing. Thus, the RIP assay was employed to assess enrichment of *U snRNAs* that comprise the major spliceosome in a SUMO modification-dependent manner (Figure 6a,b). Interestingly, our findings show that less SUMO modification of OsFKBP20-1b impaired the association with *U snRNAs* in the nucleus, along with a previous report about the Prp3 splicing factor [38]. The deficient SUMOylation of Prp3 appears to reduce the interaction of the *U2* and *U5 snRNAs*, but it has not much impact on *U4* and *U6 snRNAs*. However, with the less SUMOylated OsFKBP20-1b, the levels of *U1*, *U2*, *U4*, *U5*, and *U6 snRNAs* were prominently reduced under heat stress, suggesting that the association of the OsFKBP20-1b splicing factor with the snRNPs within the spliceosome complex is mediated by the SUMO mechanism during the stress response in plants. Further, our data demonstrate that the SUMO modification of OsFKBP20-1b is relevant for RNA processing in plant cells (Figure 6c–e). Our findings suggest that posttranslational modification of OsFKBP20-1b by SUMOylation can be a key factor that regulates the function of OsFKBP20-1b in pre-mRNA splicing upon heat stress. 

## 4. Materials and Methods

### 4.1. Subcellular Localization and Bimolecular Fluorescence Complementation (BiFC) Assay

For the localization assay, the gene constructs encoding OsFKBP20-1b and OsSCE1s or OsSUMOs were cloned into the pCAMBIA1302 binary vector, and for the BiFC assay, each construct was cloned into the pSPYNE or pSPYCE binary vector. The recombinant plasmids were used to transform *Agrobacterium tumefaciens* GV3101 cells. Suspensions of recombinant *A. tumefaciens* cells were adjusted to an optical density at 600 nm (OD_600_) of 0.5–1.0 in infiltration buffer (10 mM MgCl_2_, 10 mM MES pH 5.7, and 200 µM acetosyringone) and then used to infiltrate the leaves of 4-week-old *Nicotiana benthamiana* plants harboring the silencing suppressor p19. After two days, the tobacco plants were exposed to heat stress by incubation at 42 °C for 1 h. The leaves were observed using a confocal laser scanning microscope.

### 4.2. Reconstituted SUMOylation in E. coli

The reconstituted SUMOylation system, which has been previously reported [42], is based on *E. coli* BL21 (DE3) cells harboring the plasmids for E1 (pACYCDuet-1-His-tag-AtSAE2/AtSAE1a-S-tag) and E2 (pCDFDuet-1-His-tag-AtSUMO either with the C-terminal functional GG or mutated to alanine–alanine, AA/AtSCE1a-S-tag). Here, we inserted a construct encoding OsFKBP20-1b (complete coding sequence) C-terminally linked to the 2 × HA tag into the pET28a vector and cloned the new plasmid construct in *E. coli* BL21 (DE3) cells containing the SUMOylation enzymes. The transformed *E. coli* cells were incubated until the OD_600_ of 1.0 was reached; then, OsFKBP20-1b expression was induced by adding 0.2 mM IPTG. After 12 h incubation, cells were harvested and resuspended in PBS solution. The total soluble protein fraction was isolated and immunoblotted with primary antibody (anti-HA, Sigma-Aldrich, St. Louis, MO, USA) and secondary antibody (anti-mouse, Thermo Fisher Scientific, Waltham, MA, USA). Protein bands were visualized using the SuperSignal West Pico ECL solution (Thermo Fisher Scientific, Waltham, MA, USA) for detection with the MyECL imager system (Thermo Fisher Scientific, Waltham, MA, USA).

### 4.3. In Vitro and In Vivo Protein Stability Assays

OsFKBP20-1b-2HA protein was captured using anti-HA magnetic beads (Thermo Fisher Scientific, Waltham, MA, USA) and eluted. The total protein fraction was extracted from whole tissues of 7-day-old rice seedlings using PBS solution without protease inhibitor or protein detergent. OsFKBP20-1b-2HA protein (2 μg) and the protein extract (30 μg) were incubated with or without 50 μM MG132 for 0, 30, 60, and 120 min at 25 °C. OsFKBP20-1b_2HA protein was detected by western blot using anti-HA antibodies. The relative band intensity was calculated using the ImageJ software, and each result is provided as the value relative to the baseline at 0 min. For the in vivo protein assay, WT_ and 5KR_OsFKBP20-1b-2HA were inserted into the pSPYCE binary and pCAMBIA1302 vectors and cloned in *A. tumefaciens*. Suspensions of recombinant *A. tumefaciens* cells were used to infiltrate *N. benthamiana* leaves for transient co-expression of both recombinant proteins. Two days after infiltration, the total protein fraction was extracted from the leaves and incubated on ice for 30 min in protein lysis buffer (50 mM Tris-Cl, 100 mM NaCl, 0.1% NP-40, 1 mM DTT) without protease inhibitor or protein detergent. Each total protein fraction (20 μg) was incubated with 50 μg CHX or without 50 μM MG132 for 0, 3, and 6 h at 25 °C. OsFKBP20-1b proteins were detected by western blot using anti-HA antibodies. GFP protein was used as the expression control. The relative band intensity was calculated using ImageJ software. Each result is provided as the value relative to the baseline at 0 min.

### 4.4. Heat Stress Treatment for SUMOrylated Protein Detection 

Flag tagged WT_ and 5KR_*OsFKBP20-1b* were inserted into the pCAMBIA1390 vector and cloned in *A. tumefaciens*. Suspensions of the transformant *A. tumefaciens* cells were used to infiltrate *N. benthamiana* leaves for transient expression of both recombinant proteins. Two days after infiltration, the leaves were treated with 100 µM cyclohexamide before heat treatment at 42 °C for 0, 3, 6, and 10 h. Total protein fraction was extracted from the leaves in protein lysis buffer (137 mM NaCl, 2.7 mM KCl, 8 mM Na_2_HPO_4_, and 2 mM KH_2_PO_4_, pH 7.4) with 5× larmmni sample buffer. WT_ and 5KR_OsFKBP20-1b proteins were detected by western blot using anti-Flag and anti-SUMO1 antibodies. The relative band intensity was calculated using ImageJ software. Each result is provided as the value relative to the baseline at 0 h. Three biological repeat experiments were performed.

### 4.5. Nuclear Protein Extraction and RNA Immunoprecipitation (RIP)

RIP analysis was performed to assess RNA and protein interactions. FLAG-tag-*OsFKBP20-1b* was transiently expressed in *N. benthamiana*. Two days after infiltration, the leaves were fixed in 1% formaldehyde for 20 min under vacuum. Then, the leaves were washed 3 times with 125 mM glycine. The nuclear protein fraction was extracted as described previously [40]. Briefly, FLAG-tag-OsFKBP20-1b was transiently expressed in *N. benthamiana*. The leaves were finely homogenized using a mortar and pestle. The powder sample was suspended in nuclear lysis buffer (NLB, 20 mM Tris-HCl pH 7.4, 20 mM KCl, 2 mM EDTA, 1 mM DTT, 2.5 mM MgCl_2_, 25% glycerol, 250 mM sucrose), filtered through Miracloth (Merck Millipore, Burlington, MA, USA), and then centrifuged. The pellet was washed 7–8 times with nuclear resuspension buffer (20 mM Tris-HCl pH 7.4, 2.5 mM MgCl_2_, 25% glycerol, 0.5% Triton X-100) until the color of the pellet turned slightly white. The nuclear pellet was resuspended in NEB buffer (50 mM Tris-HCl pH 7.5, 10 mM EDTA, 1% SDS, 1 mM PMSF) containing protease inhibitor (1×), incubated on ice for 30 min, and sonicated 7 times at 20% power for 2 s. After centrifugation, the supernatant was collected and incubated with GFP trap (Chromotek, Planegg-Martinsried, Munich, Germany) for immunoprecipitation (IP). Then, the beads were washed 3–4 times with TBST (TBS and 0.1% Tween-20), and the total RNA and proteins were extracted using the beads.

### 4.6. PEG-Mediated Protoplast Transfection

Rice protoplasts were prepared from the stem part of 10-day-old seedlings. Stems were longitudinally cut 4–5 times with a razor blade and immediately immersed in an enzyme solution (1.5% Cellulase R-10, 0.75% Macerozyme R-10, 0.5 M mannitol, 10 mM MES pH 5.7, 0.1% BSA, 10 mM CaCl_2_, 5 mM mercaptoethanol) for overnight incubation at 28 °C in the dark. Then, two volumes of W5 solution (154 mM NaCl, 125 mM CaCl_2_, 5 mM KCl, 2 mM MES pH 5.7) were added to the enzyme solution, and protoplasts were recovered by filtration through a 40 μm mesh. The protoplasts were collected by centrifugation at 300× *g* for 5 min and washed 3 times with W5 solution. The protoplast pellet was resuspended in MMG solution (0.5 M Mannitol, 4 mM MES pH 5.7, 15 mM MgCl_2_), and the protoplasts were counted using a hemocytometer. A total of 2.0 × 10^5^ cells were mixed with plasmid DNA (4 μg per construct) and one volume PEG solution (0.2 M Mannitol, 100 mM CaCl_2_, 40% PEG400). The transfection mix was incubated for 20 min at room temperature in the dark. The incubation was stopped, and 2–3 volumes of W5 solution were added slowly and mixed gently. Protoplasts were recovered by centrifugation at 300× *g* for 2 min. After washing with W5 solution, the protoplasts were resuspended in WI solution (0.5 M Mannitol, 20 mM KCl, 4 mM MES pH 5.7) and incubated overnight. For protein or RNA analysis, 22 h after the incubation (0 h), total RNA was extracted before and after a heat stress treatment for 30 and 60 min, and total proteins were extracted before and after continued incubation for 10 h with or without MG132. Immunoblot analysis was performed using anti-HA, anti-FLAG, anti-GFP, and anti-tubulin antibodies. In a semi-quantitative reverse transcription PCR (RT-PCR) analysis, cDNA was synthesized from total mRNA samples and amplified by PCR to determine target transcripts levels. The relative band intensities were calculated using the ImageJ software.

### 4.7. Semi-Quantitative RT-PCR and RT-qPCR Analyses

Total RNA was extracted from transfected *osfkbp20-1b* knock out (*k/o*) protoplast or RNA was recovered from anti-FLAG magnetic beads IP samples using the RNAiso Plus kit (TAKARA, Tokyo, Japan), according to the manufacturer’s protocol. RNA was treated with DNase at 37 °C for 30 min, and cDNA was synthesized using PrimeScript Reverse Transcriptase (TAKARA, Tokyo, Japan). Semi-quantitative RT-PCR was used to determine the expression level of alternatively spliced isoforms after normalization to *OsActin1* expression level as the internal control. RT-qPCR was performed using a Bio-Rad CFX Real-Time PCR system (Bio-Rad, Hercules, CA, USA) with SYBR, according to the manufacturer’s instructions. At least three biological replicates, each with three technical replicates, were performed for all the experiments. Primers for isoform detection, including introns, and for quantifying gene expression levels, are shown in Appendix A.

## Figures and Tables

**Figure 1 ijms-22-09049-f001:**
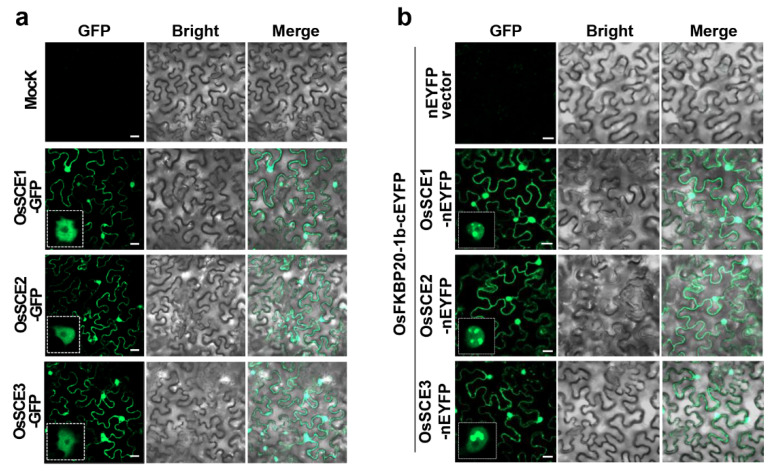
Interaction of OsFKBP20-1b with OsSCEs and the reconstituted SUMOylation assay. (**a**) Subcellular localization of OsSCE1-, OsSCE2-, and OsSCE3-GFP proteins. (**b**) BiFC assay of OsFKBP20-1b with OsSCEs. OsSCE constructs were cloned into pSPYNE (nEYFP) vector, and OsFKBP20-1b was cloned into pSPYCE (cEYFP) vector. Each construct was transformed into *A. tumefaciens* (GV3101) and infiltrated to leaves of 4-week-old *N. benthamiana*. Two days after infiltration, the leaves were observed using a confocal microscope. Bar = 50 μm.

**Figure 2 ijms-22-09049-f002:**
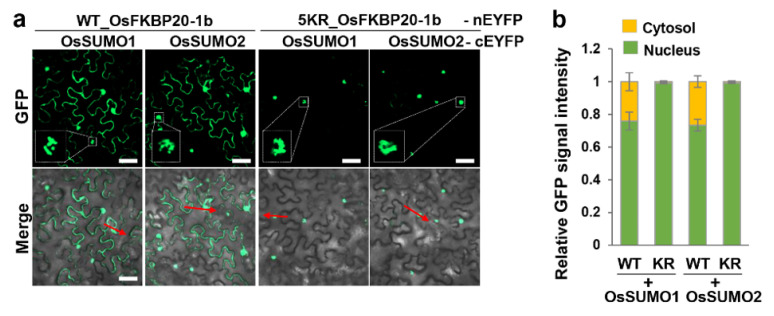
5KR_OsFKBP20-1b affects cytoplasmic interaction with SUMO proteins. (**a**,**b**) Subcellular interaction between WT_OsFKBP20-1b or 5KR_OsFKBP20-1b and OsSUMO1/2 by BiFC assay. OsSUMO1, 2 and 3 constructs were cloned into pSPYCE (cEYFP) vector and OsFKBP20-1bs were cloned into pSPYNE (nEYFP) vector. Each construct was transformed into *A. tumefaciens* (GV3101) and infiltrated to leaves of 4-week-old *N. benthamiana*. (**a**) Two days after infiltration, the leaves were observed using a confocal microscope. Bar = 50 μm. (**b**) Relative GFP signal intensity of WT or 5KR_OsFKBP20-1b interacting with OsSUMOs. The signal intensity was measured by profiling process of ZEN system. Red arrows in (**a**) represent the one of the profiling data. Error bar was standard deviation (*n* = 20 cells).

**Figure 3 ijms-22-09049-f003:**
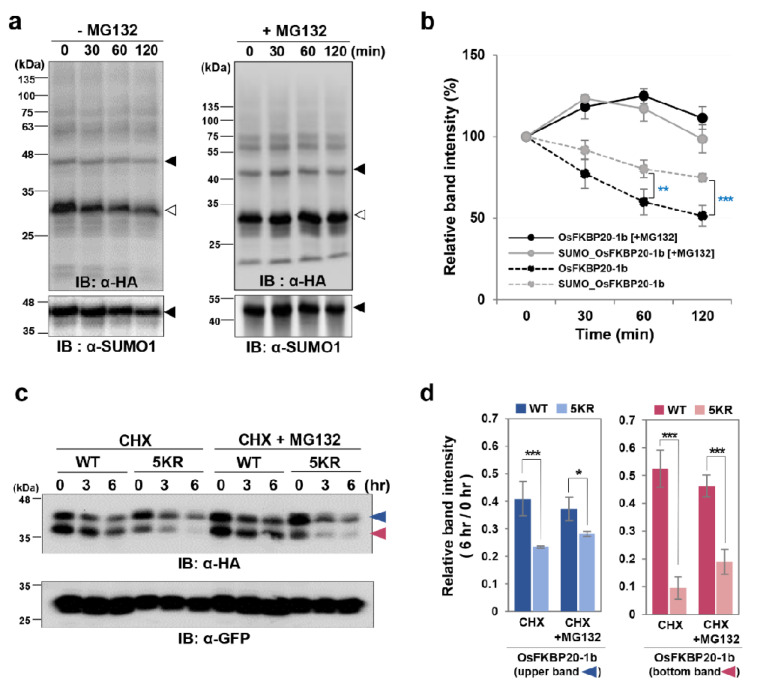
SUMOrylaed OsFKBP20-1b enables maintain the stability both in vitro and in vivo. (**a**) Degradation of recombinant WT and 5KR_OsFKBP20-1b proteins with HA tag in cell extracts of plants. Purified WT_OsFKBP20-1b and 5KR_OsFKBP20-1b proteins (2 μg) were incubated with cell extracts (30 μg) isolated from 7-day-old rice seedlings for indicated times at 25 °C with/without 50 μM MG132. OsFKBP20-1b protein levels were detected by immunoblot (IB) using anti-HA antibodies. (**b**) SUMOylated OsFKBP20-1b (black arrow head) and non-SUMOylated OsFKBP20-1b (white arrow head) bands were detected by immunoblot assay at (**a**) and were calculated using ImageJ software. The data are represented as values which were normalized to time zero (baseline). Error bar indicates ± SE of three repeated experimental data sets. (**c**) In vivo degradation of WT and 5KR_OsFKBP20-1b proteins through transient expression in *N. benthamiana* leaves. GFP and HA tagged WT or 5KR_OsFKBP20-1b protein was transiently expressed by *A. tumefaciens* infiltration. At 2 days after infiltration, total proteins extracted from tobacco leaves and treated CHX with (CHX + MG132) or without MG132 (CHX) for 0, 3 and 6 h. There are two bands for OsFKBP20-1b, ◄ indicates upper band; ◄ indicates lower band. The protein expression was detected by immunoblot assay. GFP was used for control for transient protein expression level. (**d**) Relative WT_OsFKBP20-1b and 5KR_OsFKBP20-1b degradation levels in vivo. The band intensities of WT_ and 5KR_OsFKBP20-1b at 6 h was divided to 0 h band using ImageJ program. Error bar indicates ± SE of three times repeated experimental data sets. Student’s *t*-test; * *p*-value < 0.1, ** *p*-value < 0.05, *** *p*-value < 0.01.

**Figure 4 ijms-22-09049-f004:**
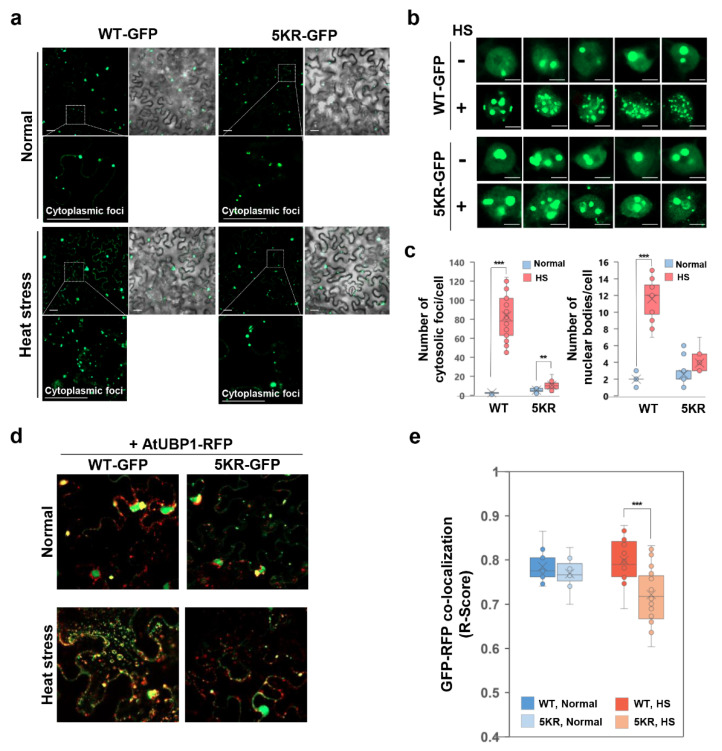
Comparison of nucleus-cytosolic foci localization between WT_OsFKBP20-1b and 5KR_OsFKBP20-1b mutant under heat stress. GFP fused to OsFKBP20-1b WT and OsFKBP20-1b 5KR mutant constructs were transiently expressed in leaves of 4-week-old *N. benthamiana* using *A. tumefaciens* infiltration. Two days after infiltration, plants were treated with heat stress on 42 °C for 1 h and then observed GFP fluorescence using confocal microscopy. (**a**,**b**) Localization of GFP fused OsFKBP20-1b WT and 5KR mutant after heat stress in epidermal cells. (**a**) Cytosolic foci focusing images; bar = 20 μm. The dotted white box indicates the magnified region show in the inset. (**b**) Nucleus focusing images; bar = 5 μm. (**c**) Number of cytoplasmic foci and nuclear bodies. The numbers of foci were totally counted and divided the number of cells. Error bars represent standard deviation (*n* = 20 cells). (**d**,**e**) Co-localization of OsFKBP20-1b WT or 5KR with stress granule. Stress granule marker protein AtUBP1-RFP and the OsFKBP20-1b-GFP constructs co-expressed in the leaves of *N. benthamiana*. GFP and RFP co-localization score was calculated using the JACoP software and is represented as R-score. Student’s *t*-test; ** *p*-value < 0.05, *** *p*-value < 0.01. WT-GFP, WT_OsFKBP20-1b-GFP; 5KR-GFP, 5KR_OsFKBP20-1b-GFP.

**Figure 5 ijms-22-09049-f005:**
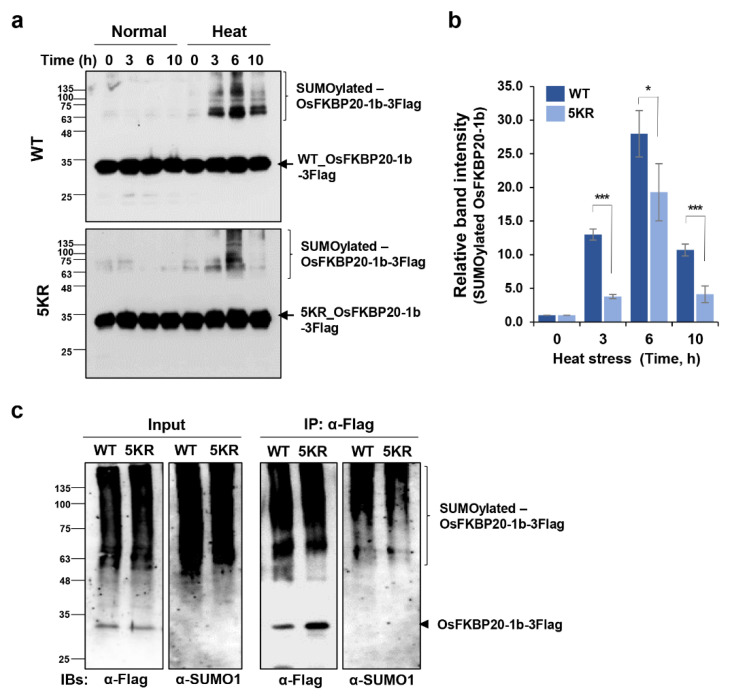
OsFKBP20-1b is SUMOylated by heat stress in planta. Flag tagged WT_ and 5KR_OsFKBP20-1b proteins were transiently expressed in the leaves of 4-week-old *N. benthamiana* using *A. tumefaciens* infiltration. Two days after infiltration, plants were treated with 100 μM CHX and heat stress at 42 °C for 0, 3, 6, 10 h. (**a**) Total proteins were extracted from the leaves and detected by anti-Flag antibody. (**b**) SUMOylated bands were calculated using imageJ program. Error bars represent standard errors from three biological repeat experiments. * *p*-value < 0.1, *** *p*-value < 0.01. (**c**) Immuno-precipitation assay of OsFKBP20-1b under heat stress. Two days after infiltration, plants were treated with 100 μM CHX and heat stress at 42 °C for 3 h. After nuclear protein fractionation, OsFKBP20-1b-Flag proteins were precipitated using anti-Flag bead and detected by anti-Flag or anti-SUMO antibody.

**Figure 6 ijms-22-09049-f006:**
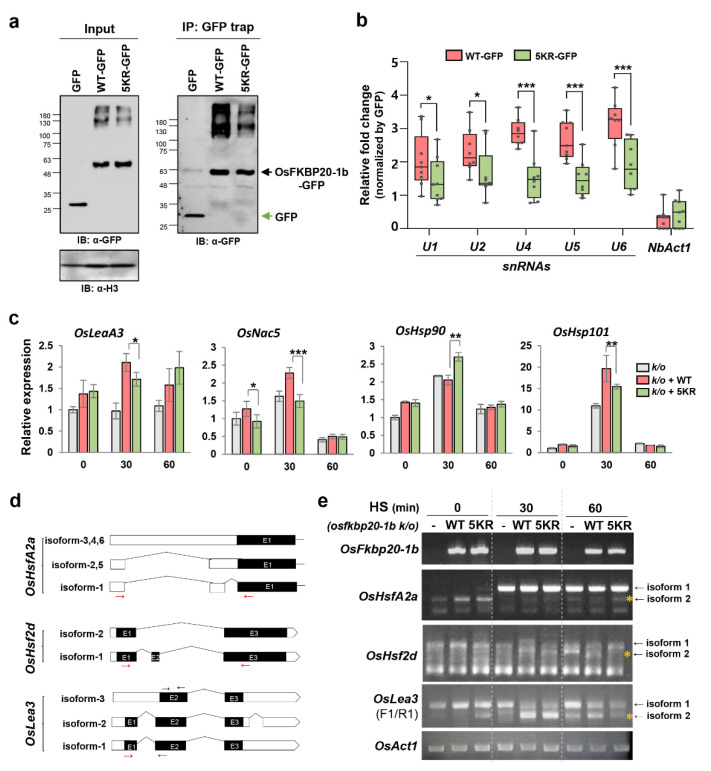
snRNAs and transcription level of stress responsive genes in association with either WT_OsFKBP20-1b or 5KR_OsFKBP20-1b. (**a**,**b**) GFP fused OsFKBP20-1b WT and OsFKBP20-1b 5KR mutant constructs were transiently expressed in leaves of *N. benthamiana* using *A. tumefaciens* infiltration. (**a**) GFP fused proteins were extracted from the leaves and these proteins were purified by immunoprecipitation (IP) using a GFP trap. HistoneH3 antibody used for nuclear protein loading control. (**b**) RNA immunoprecipitation (RIP) assay. *NbSnRNAs* (*U1*, *U2*, *U4*, *U5*, and *U6*) and *NbActin* (*NbAct1*) were detected by RT-qPCR. The value of GFP fused to *OsFKBP20-1b* WT and 5KR was normalized by GFP control. Bars represent SEM of nine biological replicates. Student’s *t*-test; * *p* < 0.1, ** *p* < 0.05, *** *p* < 0.005. (**c**–**e**) Protoplasts were isolated from the shoots of *osfkbp20-1b k/o* seedlings, and WT_ and 5KR_OsFKBP20-1b proteins were transiently expressed by PEG mediated transfection assay. Total RNA was isolated at 22 h (0 min) after protoplast transfection and 30–60 min after heat stress treatment. (**c**) Transcript expression levels of abiotic stress-responsive genes (*OsLea3* and *OsNac5*) and heat stress-responsive genes (*OsHsp90* and *OsHsp101*). Transcripts were detected using semi-quantitative RT-PCR analysis. *OsAct1* was used as the normal control. Bars represent mean SEM of three biological replicates. Student’s *t*-test; * *p* < 0.05, ** *p* < 0.01. (**d**,**e**) Semi-quantitative RT-PCR analysis of alternative splicing patterns of stress-responsive genes in mutant seedlings. Alternatively, spliced isoforms of *OsHsfA2a*, *OsHsf2d*, and *OsLea3* are presented in (**d**). *OsAct1* was used as a loading control. Red arrows indicate the primer sites used.

## Data Availability

The data represented in this study are available on request from the corresponding authors. The data are not publicly available as they are contained in laboratory notebooks.

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
