# Peer review of "SUMO Modification of OsFKBP20-1b Is Integral to Proper Pre-mRNA Splicing upon Heat Stress in Rice"

_ijms, 2021, doi:10.3390/ijms22169049_

Round 1
Reviewer 1 Report
Line 84: a brief explanation of OsFKBP20-b (any known function) will be helpful for the audience
Lines 99 – 100: please rewrite, ‘localization of rice the OsSCEs’ does not make sense.
Line 101: GFP was fused to which end (N or C terminus) of OsSCEs?
Lines 195-196: please rewrite, incomplete sentence
Line 261: exam?
Italicize scientific names and distinguish proteins from gene(s) by using capital letters for proteins
Although SUMOylated mutant assay did not provide clear conclusions, other evidence positively supported the authors’ hypothesis.
The reconstituted assay story using E. coli can move to the supplement because in planta study supported the authors’ findings.
It is highly recommended to identify the SUMO target sites of OsFKBP20-1b.
Reviewer 2 Report
I am satisfied with the corrections and changes made. Thank you.
This manuscript is a resubmission of an earlier submission. The following is a list of the peer review reports and author responses from that submission.
Round 1
Reviewer 1 Report
Line 84: a brief explanation of OsFKBP20-b (any known function) will be helpful for the audience
Lines 99 – 100: please rewrite, ‘localization of rice the OsSCEs’ does not make sense.
Line 101: GFP was fused to which end (N or C terminus) of OsSCEs?
Lines 195-196: please rewrite, incomplete sentence
Line 261: exam?
Italicize scientific names and distinguish proteins from gene(s) by using capital letters for proteins
Although SUMOylated mutant assay did not provide clear conclusions, other evidence positively supported the authors’ hypothesis.
The reconstituted assay story using E. coli can move to the supplement because in planta study supported the authors’ findings.
It is highly recommended to identify the SUMO target sites of OsFKBP20-1b.
Author Response
Thank you very much for taking the time to review our manuscript and provide valuable feedback. I have responded to each of your comments below.
- The reconstituted assay story using E. coli can move to the supplement because in planta study supported the authors’ findings.
Response: We have now moved SUMOylation assay in a reconstitutive E. coli system to the Supplementary Figure S2, as your suggestion.
- Italicize scientific names and distinguish proteins from gene(s) by using capital letters for proteins
Response: We have now revised the gene and protein names according to your comment. Gene name is to italicize with the lowercase letter except the first letter, whereas as symbols for proteins are not italicized with capital letters.
- Line 84: a brief explanation of OsFKBP20-b (any known function) will be helpful for the audience
Response: I have revised the sentence with a brief explanation of OsFKBP20-1b function in line 84-85.
- 4. Lines 99 – 100: please rewrite, ‘localization of rice the OsSCEs’ does not make sense.
Response: I have corrected the sentence to “the subcellular localization of the rice OsSCEs” in line 103.
- Line 101: GFP was fused to which end (N or C terminus) of OsSCEs?
Response: I have revised this sentence as follows: “all three C-terminal GFP fusion OsSCE (OsSCE1, 2, 3-GFP) proteins were~” in line 103-104, as per your comment.
- Lines 195-196: please rewrite, incomplete sentence
Response: I have corrected the sentence as follows: “~ neither the SUMOylated nor the non-SUMOylated OsFKBP20-1b protein was not degraded throughout the time course experiment (Figure 3a)” in line 201, as per your comment.
- Line 261: exam?
Response: I have changed the verb “exam” to investigate for the audience in line 268.
Reviewer 2 Report
The work describes FKBP20-1b of rice to be sumoylated at multiple Lys (K) sites in order to enhance its stability. FKBP20-1b interacts with sumo1 and 2, but also with spiceosome U1-5 RNA components. Low sumoylation levels of FKBP20-1b led to inaccurate splicing and transcription under heat stress conditions. It seems that sumoylation also alters sub-cellular localisation of the protein.
- During HS sumoylation of many proteins is increase in order to regulate their stability and hamper aggregation (to aid latter refolding or ubiquitination and decay). The authors should analyse the overall level of FKBP20-1b by western blot; it is expected that it will increase during HS since sumoylation leads to its stability.
- As FKBP20-1b increases expression of HSP genes (70, 90, 101), and contributes to efficient splicing of many transcripts, it is expected that will enhance heat tolerance: please test heat tolerance during various HS regimes ()BT, SAT, LAT etc.) by comparing wt, fkbp20-1b k/o and complemented lines (WT and 5KR constructs). I understand these lines are available.
Author Response
Thank you very much for taking the time to review our manuscript and provide valuable feedback. We have conducted experiments as requested by yourself and responded to each comment below.
- During HS sumoylation of many proteins is increase in order to regulate their stability and hamper aggregation (to aid latter refolding or ubiquitination and decay). The authors should analyse the overall level of FKBP20-1b by western blot; it is expected that it will increase during HS since sumoylation leads to its stability.
Response: We have revised Supplementary Figure S4 adding the total protein amount analysis of OsFKBP20-1b between normal and heat stress conditions. As shown in Supplementary Figure S4a, OsFKBP20-1b protein expression under 35S promoter was not increased by heat stress, on the contrary it was rather a little decreased upon heat stress. The reason is that the cytosolic protein which make up the majority of the protein is prominently decreased, whereas the nuclear protein amount is increased under heat stress in Supplementary Figure S4b and c in lines 288-289 and 292-294. Therefore, further studies are needed to understand the molecular mechanism of OsFKBP20-1b stability by SUMO regulation.
- As FKBP20-1b increases expression of HSP genes (70, 90, 101), and contributes to efficient splicing of many transcripts, it is expected that will enhance heat tolerance: please test heat tolerance during various HS regimes (BT, SAT, LAT etc.) by comparing wt, fkbp20-1b k/o and complemented lines (WT and 5KR constructs). I understand these lines are available.
Response: Although, a sufficient number of knock out plants (osfkbp20-1b k/o) was not used for heat stress phenotyping, in the previous experiment we found that dysfunction of OsFKBP20-1b plants were more sensitive to heat stress than the wild type plants in Supplementary Figure S8 and in line 382-388. Unfortunately, we did not generate the complemented lines by WT_ and 5KR_OsFKBP20-1b yet.
Round 2
Reviewer 2 Report
Thank you for the efforts you have made. After revision, I believe that there is not enough data to support the conclusions. Specifically, the authors state that sumoylation increases protein stability, however this is not entirely clear (it rather changes sub-cellular localisation): (i) in S4b figure the cytoplasmic amount of FKBP20-1b is decreased by heat in both wt and 5KR mutant is about the same extent, although based on quantification the 5KR protein is not further down-regulated by heat: importantly on the western blot proteins in lane 2 and 4 look clearly different; (ii) S4b quantification has been done on technical reps and not bio ones!!; (iii) if sumoylation increases stability, why we cannot see the increased sumoylated form in the nuclear fraction (Fig S4b, left panel); (iv) it is unclear how 5KR affects sumoylation: based on Fig 2c anti-Flag IP samples of sumo-5KR are increased relative to sumo-wt, while are decreased when analysed by anti-sumo. It is possible that 5KR mutations affect other features of protein but not sumoylation.
I apologise if I misunderstood anything.